# Married and Cohabiting Finnish First-Time Parents: Differences in Wellbeing, Social Support and Infant Health

Mirjam Kalland [1,*], Saara Salo [1], Laszlo Vincze [2], Jari Lipsanen [3], Simo Raittila [4], Johanna Sourander [1], Martina Salvén-Bodin [1] and Marjaterttu Pajulo [5]

1 Faculty of Educational Sciences, Siltavuorenpenger 5, P.O. Box 9, University of Helsinki, 00014 Helsinki, Finland; saara.z.salo@helsinki.fi (S.S.); johanna.sourander@helsinki.fi (J.S.); martina.salven@helsinki.fi (M.S.-B.)
2 Swedish School of Social Sciences, Snellmaninkatu 12, P.O. Box 16, University of Helsinki, 00014 Helsinki, Finland; laszlo.vincze@helsinki.fi
3 Faculty of Medicine, Haartmaninkatu 8, P.O. Box 63, University of Helsinki, 00014 Helsinki, Finland; jari.lipsanen@helsinki.fi
4 Faculty of Social Sciences, Unioninkatu 33, P.O. Box 42, University of Helsinki, 00014 Helsinki, Finland; simo.raittila@helsinki.fi
5 Department of Clinical Medicine, University of Turku, 20014 Turku, Finland; marpaj@utu.fi
* Correspondence: mirjam.kalland@helsinki.fi

**Abstract:** Cohabitation is more common than marriage when couples are expecting their first child in Finland. However, little is known about possible differences in wellbeing between the two groups. In this study, we examined differences in parental wellbeing, relationship satisfaction, infant health outcomes, and use of social support among cohabiting and married first-time parents. Survey data was collected from 903 parents during pregnancy and at one month postpartum. Cohabiting parents had more depressive symptoms than married parents. They were also less satisfied with their relationships and expressed less satisfaction with the quality of support they got from their partner. Cohabiting fathers did not use the cost-free support from maternity clinics as often as married fathers. Our results show differences in well-being between married and cohabiting first-time parents and that the support from maternity clinics should better acknowledge diversity and address the different needs of different types of families.

**Keywords:** family formation; first-time parents; cohabitation; marriage; depression; relationship satisfaction; support





## 1. Introduction

Cohabitation versus marriage was a 'hot topic' at the end of the last century and in the first decade of this century, both in sociology (Giddens 1992; Beck and Beck-Gersheim 1995) and in research on family formation as a basis for child development (Brown 2004, 2010; Cavanaugh and Huston 2006; Manning and Brown 2006). British demographer Kathleen Kiernan (2004) noted at the time that cohabitation was being accepted in stages in European nations, and that the Nordic countries in particular were further along these stages than the others. According to Kiernan, in the first stage, cohabitation is a fringe or avantgarde phenomenon. In stage two, cohabitation is accepted as a testing ground for marriage. In the next stage, it becomes an alternative to marriage, and finally, in stage four, it becomes indistinguishable from marriage. In an introduction (2004) of his concept of the 'deinstitutionalization of marriage', sociologist Andrew Cherlin discussed contemporary family change and asked why so many people are marrying, planning to marry, or hoping to marry, when cohabitation and single parenthood are widely acceptable options. He argued that although the practical importance of being married has declined, its symbolic importance may have even increased. Indeed, whereas marriage is often celebrated, for

many couples, moving in together seems to be a rather ambiguous commitment 'fuelled by pragmatism rather than romance' (Lindsay 2000).

In general, earlier studies indicate that marriage is better for both relationship stability and child outcomes (Manning and Brown 2006; Gibson-Davis and Brooks-Gunn 2007; Guzzo and Lee 2008; Schmeer 2011; Shah et al. 2011). However, the results might reflect that cohabiting couples differ from married in many ways, such as socioeconomic background or relationship commitment (Brown 2010). With regard to relationship satisfaction, it has been suggested that in countries that lack a conjunctive norm against childbearing in cohabiting unions, such as Finland, cohabiting as parents has no negative effect on life-satisfaction (Stavrova and Fetchenhauer 2015).

In one study, which assessed pregnancy outcomes in the 1990s in Finland (Raatikainen et al. 2005), the hypothesis was that after controlling for confounding factors, little or no difference would remain between the pregnancy outcomes of the study groups. However, the study found that pregnancy outside marriage was associated with an overall 20% increase in adverse pregnancy outcomes such as low birthweight or preterm delivery. Being unmarried was also associated with lower socioeconomic status. According to an annual review by Statistics Finland (OSF 2016b), 57% of first-borns were born outside marriage in Finland in 2015, whereas the percentage was 38 in 1990. Of second or higher order children, 35% were born outside marriage.

However, even though cohabiting is more common than marriage when couples are expecting their first child, research in Finland has shown little interest in comparing those two groups of parents and their children. This is surprising, especially as cohabiting first-time mothers are at a greater risk of splitting up (Airio 2016), which might have an impact on child outcomes. Thus, little is known about whether Finnish expecting, cohabiting, first-time parents differ from married parents in their terms of their wellbeing and health behaviours during pregnancy and post-birth, especially when socioeconomic status is controlled for. In addition, little is known about whether cohabiting expecting parents are more dissatisfied with their relationships than expecting married parents. It seems that, as a majority of expecting first-time mothers are non-married, it has been taken for granted that this change has little or no impact on the wellbeing of the parents or on the health of the child. In this study, we aim to explore what has been taken for granted, focusing not only on health and wellbeing but also on the use of social support among expecting first-time parents.

In this paper, we focus on the following aspects of maternal and paternal health related to marital status during pregnancy, which have important implications for child development: (1) maternal and paternal depression during pregnancy, (2) maternal and paternal relationship satisfaction, (3) baby being planned for, as a possible variable related to relationship commitment, (4) foetal exposure to cigarette smoke, and (5) access to and use of support during pregnancy. In addition, we focus on the health of the new-born child.

The rationale for focusing on these aspects lies in the existing literature. Parental depression, especially among mothers, is widely recognised as being associated with various later negative developmental outcomes in the child (Field 2011). Relationship satisfaction and quality is related to child outcomes, as children of mothers in unstable relationships show a higher probability of suffering from emotional or behavioural problems than children of mothers with high relationship quality (Hannighofer et al. 2017). Smoking during pregnancy is associated with both short-term negative outcomes such as preterm delivery and low birthweight (Bell et al. 2018; Shah and Bracken 2000) with motherhood at risk (Kalland et al. 2006) and long-term outcomes such as negative temperament in early childhood (Brook et al. 2000) and problems with cognition and behaviour (Ernst et al. 2001; Najman et al. 2004; Rodriguez and Bohlin 2005).

Finally, social support is considered important for buffering against or alleviating the negative effects of stress on women's physical and mental health, and, therefore, for protecting maternal and foetal wellbeing during pregnancy (Young et al. 2019). In Finland, pregnant mothers and their partners are provided free-of-charge services during pregnancy

in maternity clinics. These services are universal, and according to national statistics, 99.7% of mothers use them during pregnancy. Of high-risk mothers, as many as 97.8% use the services (Kalland et al. 2006). However, to our knowledge, no previous research has examined how the use and perception of maternity clinics differ in terms of marital status for expecting parents. When expecting their first child, the partner is considered an important source of social support for the mother (Tanner Stapleton and Bradbury 2012). The quality of this support varies greatly, depending on many factors within and outside the relationship. In addition, the social network of new parents comprises informal support—from family or friends; and formal support—in the form of services from professionals (Leahy-Warren 2016) or through the internet and social media (Baker and Yang 2018).

The current study is part of a larger study of first-time parents in Finland, Families First study. The study follows 1005 first-time parents-to-be from pregnancy until the child is about two years old, in five waves, with an option for later follow-ups (permission requested from the parents and obtained from 403 parents). The scientific goal of the study is to provide empirical data on how the development process of becoming a parent (mother or father) in Finland today is affected by previous experiences in childhood, by relationship status and quality, and by social support. In this study, we investigate the baseline and first month after delivery (wave one and two) in terms of impact of marital status on relationship satisfaction, health behaviour, depression, infant health and perceived support among this sample of Finnish first-time parents. Specifically, we ask:

RO1. Is marital status associated with depression, relationship satisfaction, baby being planned for, smoking, and infant outcome at one month among parents expecting their first child?

RO2. Is marital status related to how expectant parents report being supported by their partners, friends, family, and maternity clinics?

## 2. Materials and Methods

Pregnant first-time parents from 80 municipalities all over the country were invited to take part in the study (Kalland et al. 2015). Parents from 683 families (663 mothers and 350 fathers) took part in at least one phase of the study. At baseline, we received answers from 903 parents, of whom 601 were mothers and 302 fathers, and one month after delivery, 847 parents took part in the survey, of whom 572 were mothers and 269 fathers.

Data collection was carried out using online questionnaires, accessible using a personal code. Paper versions of the questionnaires were also available upon request. The public health nurses in the maternity clinics informed potential participants of the study during a regular visit in the third trimester of pregnancy. The nurses were specifically instructed not to select specific families for the study, but to offer all first-time parents the opportunity to take part in the study, during a regular check-up appointment. The parents received written information regarding the study as well as forms for informed consent together with a prepaid envelope. The data were collected during years 2012–2016.

At this point in the study, we are investigating the baseline and the first month after the delivery. The first assessment (baseline) was during the third trimester, and the second assessment was performed when the baby was one month old. The average gestational week during which the mothers answered the survey was 34.4 (SD 3.2). The mothers and fathers in the study answered the same questionnaires.

To answer the research questions, we focused on the following background variables: age, marital status, gender, and educational level.

For marital status we gave the alternatives single, cohabiting, married, divorced, widowed, other—describe. For education we used the categories described in Table 1 to be able to compare our data with national data and research reports. For smoking, we offered the parents the following possibilities: 1. Not smoking, 2. Smoked but stopped during the first trimester, 3. Smoked longer than first trimester, 4. Smoked irregularly during pregnancy. For regression analysis, the smoking categories were combined to non-smoking versus smoking (all indications of smoking during pregnancy).

**Table 1.** Mothers' Educational Level Compared to National Average.

| Educational Attainment | Mothers in Our Sample | Finnish First-Time Mothers [c] |
|:---:|:---:|:---:|
| Basic education | 18 (3.0%) | 14% |
| Second degree | 133 (22.1%) | 39% |
| Vocational college | 26 (4.3%) | 4% |
| Lower university [a] | 248 (41.3%) | 25% |
| Higher university [b] | 153 (25.5%) | 17% |
| Research education | 23 (3.8%) | 1% |

[a] Includes polytechnic. [b] Includes applied sciences. [c] From (Haataja 2014), percentages are from 2009 data.

To screen for depressive symptoms at baseline, we used the Edinburg Postnatal Depression Scale (Cox et al. 1987), which is a 10-item questionnaire originally developed to assist in identifying possible symptoms of depression in the postnatal period. It also has adequate sensitivity and specificity to identify depressive symptoms in the antenatal period and is useful for identifying symptoms of anxiety. In this study, we used it for both men and women, during both pregnancy and postpartum. In the Finnish maternity and well-baby clinics, a cut-off score of 13 is used for probable major depression, and a cut-off score of 10 for probable depression (Hakulinen and Solantaus 2016). In this study, we use the national recommendations.

For assessing relationship satisfaction, we used the Index of Marital Satisfaction (IMS) (Hudson 1997). The IMS is a 25-item scale designed to measure the degree of dissatisfaction of people with their marriage. The item scores range from 1 to 7: the higher the score, the higher the experienced dissatisfaction with the relationship. The IMS has shown to have good internal consistency (alpha 0.96) and concurrent, construct and discrimination validity (Hudson 1997). In our study, we used the scale for both married and cohabiting couples. The cut-off score, indicating problems in the relationship, is 25 for men and 28 for women. A score of 70 or over indicates serious problems in the relationship.

For assessing baby health, we asked for the child's birth weight, Apgar score (first minute) at birth, visits to health professionals due to illness or disease, medication prescribed for the baby, serious illness diagnosed before birth, and mother's and father's subjective appraisals of their baby's health. For the subjective appraisal of baby's health, the parents answered a five-point Likert scale from bad to very good. For the child's health, we relied on the mothers' answers, but if these were unavailable, we used the fathers' answers. We also asked if the baby was planned for, with the dichotomy alternative yes/no.

Finally, we asked about the support the parents had used during pregnancy. For assessing the support, we constructed a series of questions about its use (yes/no) and asked the parents to report how helpful the support had been on a Likert scale from 1 (not true at all) to 5 (completely true). The items are listed in Appendix A.

### 3. Results

We were not able to make a reliable estimation of response rates as the nurses did not keep statistics of how many envelopes they distributed to expecting parents. Therefore, to study whether the research sample at baseline is representative of first-time parents in Finland in terms of distribution of marital status, age, and educational level, we compared the demographic composition of our sample with data on first-time parents from Statistics Finland and from research publications.

According to Statistics Finland, in 2015 in Finland, the average age of first-time mothers was 28.8 years and of fathers 31.0 years (OSF 2016a). The age difference of first-time parents in Finland has remained almost unchanged for at least 30 years, and first-time fathers are slightly over two years older than first-time mothers. In our sample, the mothers' average age at delivery was 29.3 years. The difference is statistically significant, but the effect size is small $t(513) = 2.424$, $p = 0.016$. The mean age for fathers in our sample was 31.3 years and did not differ from the national average, $t(224) = 0.827$, $p = 0.409$. Tables 1 and 2 show

that both mothers and fathers in this study had a higher level of education than expected, based on the national average for first-time parents. Of the expectant mothers, 52% were married in this sample, which differs from national numbers for first-borns (57% born outside marriage in Finland in 2015).

**Table 2.** Fathers' Educational Level Compared to National Average.

| Educational Attainment | Fathers in Our Sample | Finnish First-Time Fathers [a] |
|---|---|---|
| Lower secondary | 1 (0.4%) | 17.9% |
| Upper secondary | 101 (36.2%) | 49% |
| Bachelor's | 98 (35.1%) | 20.3% |
| Master's/Doctoral | 79 (28.3%) | 12.8% |

[a] From (Paavilainen et al. 2016), percentages are from the years 2005–2009.

### 3.1. Differences in Parental Wellbeing and Infant Outcome

To answer the first research question, we first performed an OLS regression with IMS and EPDS as dependent variables (see Table 3). With respect to IMS, the model was significant and explained 3% of the variance in the dependent variable, $R^2 = 0.03$, $F(4, 757) = 6.00$, $p < 0.001$. Marital status had a significant effect indicating that relationship dissatisfaction is more typical among cohabiting parents than among married parents; gender, age, and level of education had no significant effects. The regression model was significant also regarding EPDS, and it explained 8% of the variance in the dependent variable, $R^2 = 0.08$, $F(4, 686) = 15.31$, $p < 0.001$. All four independent variables had a significant effect, and it was shown that depressive symptoms are more prevailing among cohabiting than among married parents, among mothers than among fathers, among parents with lower education than among parents with higher education; also, depressive symptoms were increasing with the decrease of age.

**Table 3.** Results from OLS regressions predicting EPDS and IMS.

|  | EPDS | IMS |
|---|---|---|
| Marital status | 0.09 * | 0.17 ** |
| Age | −0.09 * | 0.05 |
| Education | −0.11 ** | 0.04 |
| Gender | −0.22 ** | −0.03 |
| $R^2$ | 0.08 | 0.03 |
| $F$ | 15.31 *** | 6.00 *** |

*Note.* The table shows standardized regression coefficients. * $p < 0.05$, ** $p < 0.01$, *** $p < 0.001$.

Next, logistic regressions were carried out to explain smoking during pregnancy and whether the baby was planned for, or if the pregnancy was unintended (see Table 4). With respect to smoking, the regression was significant, $\chi^2 (4) = 70.69$, $p < 0.001$, and according to the Nagelkerke pseudo $R^2$, explained about 20% of the variation in the dependent variable. Specifically, marital status was significant showing that cohabiting parents had about a 2.1 times greater odds ratio for report smoking than the married parents. Age was significant, as each year of advancing age decreased the odds of smoking by about 8%. Education also had a significant impact, demonstrating that the parents with lower education had a 3.7 times greater odds ratio for reporting smoking during the pregnancy than those with higher education. Finally, gender had a significant effect demonstrating that fathers were about 3.4 times more likely to report smoking during pregnancy than mothers.

**Table 4.** Results from logistic regressions predicting Baby PLANNED for and SMOKING.

| | | PLANNED | SMOKING [1] |
|---|---|---|---|
| Marital status | cohabiting | 0.29 ** | 2.10 ** |
| | married (ref) | 1.00 | 1.00 |
| | Age | 1.06 | 0.92 ** |
| Education | higher | 1.70 | 0.27 *** |
| | lower (ref) | 1.00 | 1.00 |
| Gender | father | 1.09 | 3.40 *** |
| | mother (ref) | 1.00 | 1.00 |
| Nagelkerke $R^2$ | | 0.10 | 0.20 |
| $\chi^2$ | | 29.16 *** | 70.69 *** |

*Note.* The table shows odds ratios. [1] non-smoking used as the reference category, ** $p < 0.01$, *** $p < 0.001$.

In addition, we analysed the prevalence of depressive symptoms (10 points or more) using the EPDS. The results showed differences between the cohabiting (n = 279) and married mothers (n = 303), 12.7% and 6.9%, respectively. Differences in the prevalence of depressive symptoms were also observed between the cohabiting (n = 112) and married fathers n = 161), 5.7% and 4.1%, respectively. As regards to probable major depressive symptoms (13 points or more), we observed 7.7% among the cohabiting mothers, and 3.3% among the married mothers, but no differences between the married and cohabiting fathers.

Finally, there were no statistical differences in birth weight, gestational weeks, one-minute Apgar points, or subjective appraisal of infant health as reported by the parents (Table 5).

**Table 5.** Infant Outcome by Marital Status.

| | Cohabiting | | Married | | |
|---|---|---|---|---|---|
| | *N* | *M (SD)* | *N* | *M (SD)* | *p* |
| GESTATIONAL WEEKS [a] | 255 | 39.70 (1.81) | 282 | 39.74 (1.71) | 0.772 |
| BIRTH WEIGHT [a] | 255 | 3401 (494) | 282 | 3482 (496) | 0.059 |
| APGAR (1 min) [a] | 254 | 8.54 (1.48) | 281 | 8.54 (1.47) | 0.985 |
| SUBJECTIVE [a] | 255 | 1.21 (0.44) | 283 | 1.25 (0.47) | 0.319 |
| | *N* | *%* | *N* | *%* | |
| ILLNESS [b] | 275 | 1.4% | 303 | 0.0% | - |
| VISIT [b] | 234 | 34.6% | 281 | 35.6% | 0.818 |
| MEDICATION [b] | 238 | 5.9% | 281 | 5.3% | 0.788 |

SUBJECTIVE = parent's subjective appraisal of baby's health, ILLNESS = whether the baby was diagnosed with a serious illness before birth, VISIT = whether there has been a need to visit a healthcare professional in the first three months, MEDICATION = whether the child has been prescribed medication for an illness during the first three months. [a] For GESTATIONAL WEEKS, BIRTH WEIGHT, APGAR, and SUBJECTIVE this is the result of a *t*-test. [b] For ILLNESS, VISIT, and MEDICATION it is the significance from a $\chi^2$ test.

### 3.2. Differences in Social Support

To answer the second study question, we first checked the proportion of those who had or had not used any specific support. Then, we reported the mean values for those who had used a specific form of support so that the means showed how helpful a specific form of support was perceived as being. The Likert scales were coded as 1 (*not true at all*) to 5 (*completely true*), with an option for the question not being relevant or if such support was not used at all. A set of *t*-tests were used to prevent the missing cases from not adding up.

As already described in the Introduction, in Finland, pregnant mothers and their partners are provided free-of-charge services during pregnancy in maternity clinics. Health check-ups emphasize health guidance and advice, which are based on the family's need for support as well as issues raised by the parents. According to legislation, the maternity clinics should also offer first-time parents parental group activities during pregnancy

(Government Decree 338/2011). In Table 6a,b, we present significant differences in the use or rating of support by marital status for mothers or fathers. Almost all the mothers, regardless of whether they were cohabiting or married, used the services of maternity clinics, and there was no difference in how they rated the support. However, only 83.9% of the cohabiting fathers used the services of maternity clinics, in comparison to 95.7% of the married fathers ($p = 0.028$, $\varphi = 0.16$). In addition, 82.4% of the cohabiting fathers attended to the pregnancy groups, in comparison to 91.4% of the married fathers ($p = 0.042$, $\varphi = 0.13$). Nevertheless, the cohabiting and married fathers' rating of the service did not differ.

**Table 6.** (**a**). Use and rating of support, cohabiting and married fathers. (**b**) Use and rating of support, cohabiting and married mothers.

| (a) | | | | | | |
|---|---|---|---|---|---|---|
| | **Cohabiting Fathers** | | **Married Fathers** | | | |
| | **M(SD)** | **%** | **M(SD)** | **%** | *p*-**Value for *t*-Test** | *p*-**Value for χ²** |
| Maternity clinic | 4.08 (1.00) | 83.9% | 4.2 (0.84) | 95.7% | 0.323 | 0.028 [a] |
| Pregnancy group | 3.55 (1.18) | 82.4% | 3.68 (1.05) | 91.4% | 0.415 | 0.042 |
| Support from partner | 4.56 (0.81) | 96.7% | 4.77 (0.6) | 97.8% | 0.034 | [b] |
| Support from friends | 4.08 (0.93) | 91.2% | 4.02 (0.98) | 94.2% | 0.610 | 0.535 |
| (b) | | | | | | |
| | **Cohabiting Mothers** | | **Married Mothers** | | | |
| | **M(SD)** | **%** | **M(SD)** | **%** | *p*-**Value for *t*-Test** | *p*-**Value for χ²** |
| Maternity clinic | 4.45 (0.82) | 99.2% | 4.54 (0.72) | 98.5% | 0.208 | [b] |
| Pregnancy group | 3.55 (1.10) | 86.9% | 3.64 (1.05) | 88.9% | 0.382 | 0.467 |
| Support from partner | 4.77 (0.59) | 98.8% | 4.9 (0.36) | 99.6% | 0.004 | [b] |
| Support from friends | 4.57 (0.75) | 98.4% | 4.71 (0.61) | 98.9% | 0.016 | [b] |

[a] Yates' Correction, [b] Significance not reported due to too small (<5) expected count in multiple cells.

With regard to support from friends, we found that nearly all the mothers (98.4% vs. 98.9%) and most of the fathers (91.2% vs. 94.2%, $p = 0.405$, $\varphi = 0.06$) received support from friends. The married mothers rated support from friends and family the most highly, at 4.71 (*SD* 0.61), significantly more highly ($t(476) = 2.421$, $p < 0.05$, $\eta^2 = 0.01$) than the cohabiting mothers', who rated it as 4.57 (*SD* 0.75). Among the fathers, there was no significant difference in the ratings. Support from siblings and one's parents was received and rated by the parents equally, regardless of marital status.

Support from one's partner was nearly universally reported. However, the ratings for partner support were higher among the married than among the cohabiting parents (4.9 vs. 4.77, $t(401) = 2.913$, $p = 0.004$, $\eta^2 = 0.02$ for the mothers and 4.77 vs. 4.56, $t(146) = 2.137$, $p = 0.034$, $\eta^2 = 0.02$ for the fathers).

In this study, we observed no significant difference between the cohabiting and married mothers' use of online support in general, Facebook in particular, and reading or chatting on online boards. The same was true of the fathers, with the exception that Facebook was used by a larger share of the cohabiting fathers than the married fathers. In the use of more traditional mediums such as books, magazines, television and radio, no differences were observed. The mothers' ratings of self-sought online support did not differ, with the exception that the cohabiting women found chatting on online boards less beneficial than the married mothers. The married fathers rated reading online boards significantly more highly than the married fathers. (See Appendix A for full description of use of support by Marital Status).

## 4. Discussion

Although cohabiting as a basis for parenting is culturally widely accepted in Finland, and the majority of first-born children are born to cohabiting parents, we found differences between the wellbeing of the cohabiting and the married expecting first-time parents. The results also revealed some important differences between how different forms of social support are received and appreciated by cohabiting and married parents.

In particular, in this study, the cohabiting parents expressed more relationship dissatisfaction than the married parents and were at a greater risk of depressive symptoms. In addition, the cohabiting parents more often reported that the baby was not planned. The cohabiting parents had a lower education in this sample, and the cohabiting mothers were younger than the married mothers. However, after adjusting for educational level and age, marital status was still associated with depressive symptoms, relationship satisfaction, and the baby being planned.

We did not observe effects on pregnancy outcomes and infant health, which might be related to our sample representing higher levels of education and slightly older age than general numbers for first-time parents in Finland. As there was a small tendency for lower birth weight and more illness among infants born to cohabiting parents, it is possible that in a representative and larger sample some differences in infant health would have been observed, thus corresponding with consistent results on outcomes of maternal marital status on infant health (Shah et al. 2011).

In the revision (2020) of his concept of the 'deinstitutionalization of marriage', sociologist Andrew Cherlin further discusses contemporary family change. He argues that marriage has become a choice rather than a necessity, and that the roles within marriage must be negotiated rather than taken for granted. For many, marriage might represent the final capstone, the last piece to put in place when everything else (education, job, housing) is achieved. Although the changes in family life have in many ways contributed to an individualised way of forming relationships rather than adapting to a cultural norm, they have also contributed to more awareness of the risks related to relationships.

Ulrich Beck also points out the increasing level of 'risk-consciousness' in intimate relationships—when people see that a majority of marriages end in divorce, they might become less willing to get married (Beck and Beck-Gersheim 1995). This is not only a matter of freedom of individual choice—people are 'reflexive'—they look at society, see the changes, and make personal conclusions. Thus, decisions are not only individual but systemic: people observe what is going on and try to relate it to their own life situations. Moreover, choices are not fixed but liquid (in a liquid society "lasting gives way to the transient", see Palese 2013 and Bauman 2005) and individuals construct, adjust, or dissolve the union(s) they form with others. On the other hand, the fact that marriages are now entered into voluntarily might also reflect a stronger commitment to that particular relationship in comparison with previous marriages.

From a psychological point of view, such changes in society will inevitably have effects on intimate relationships. Research has consistently demonstrated that pregnancy causes profound psychological changes and is often a stressful event for first-time parents-to-be. The expectant non-married mothers in this study were particularly more vulnerable in terms of depressive symptoms and relationship dissatisfaction. This may be related to the psychology of pregnancy, which is a time for identity change, feelings of vulnerability, and a growing need for support and stability (Glover and Capron 2017). A systematic review of risk factors for depression during pregnancy (Lancaster et al. 2010) highlights several important correlates, such as life stress, prior depression, lack of partner's support, and relationship factors.

Based on the commitment theory framework, Stanley, Rhodes, and Markman proposed (Stanley et al. 2006) that premarital cohabitation is negatively related to the long-term course of relationships, as a subset of cohabiting partners end up together because of inertia (being constrained from breaking up) rather than as a result of productive mate selection. Following this rationale, some of the cohabiting couples in our sample might be

romantically uncommitted couples, who are prevented from breaking up because of the pregnancy, whereas the married couples in our samples might have made a strong, personal commitment not dictated by society, but from personal choice. In the end, marriage would be considered a 'safe haven' during pregnancy, less vulnerable and less unstable than a 'common law' relationship, and, thus, contributing to relationship satisfaction and mental health. A reasonable explanation for the observed differences could, thus, be that more committed couples are more likely to marry, and stronger commitment is reflected as better wellbeing during pregnancy. This conclusion is also supported by a study in Norway, which found that after controlling for various covariates, cohabitation continued to have a negative effect on relationship satisfaction (Mortensen et al. 2012).

During pregnancy, psychological health and wellbeing undergoes fluctuations across the different stages of gestation, with a higher risk of depression in the first and third trimester (Lee et al. 2007; Newham and Martin 2013). In this study, the level of depression was measured during the third trimester, and the women preparing for the birth of the child may have underlined the importance of stability and support.

The results also revealed some important differences between how the cohabiting and married parents received and appreciated support. The cohabiting parents expressed less satisfaction with the quality of support they received from their partner. In addition, the results reflected that cohabiting fathers used the support provided by maternity clinics less often, whereas the expecting mothers used the support regardless of marital status. It may be that among fathers-to-be, being married translates into being more ready to be involved in the services of maternity clinics, while unmarried fathers might be less ready for the required identity change, especially if the baby was not planned for. For mothers, for obvious physical reasons, using the support of maternity clinics may come more naturally. On the other hand, the cohabiting fathers had lower educational level than married fathers, which might be reflected in feeling less comfortable with using the service provided at the maternity clinics, as the nurses might express a middle class "habitus" (Collyer et al. 2015; Bourdieu 1978).

The main weakness of this study is its community sample, which, although gathered from different parts of Finland by public health care nurses during regular visits to maternity clinics, is not representative of first-time parents in Finland. We were not able to make a reliable estimation of response rates. Thus, the results cannot be generalised as being representative of all first-time parents in Finland as such. The main strength of the study, in turn, is that we are not studying any particular risk group, but the transition to parenthood among normative first-time parents in Finland. Another strength is that the data are gathered from both mothers and fathers.

The results of this study give some support to earlier notions about the relationship between marriage and the wellbeing of family members. This study also reflects the fact that even in a country in which the majority of first-borns are born outside marriage, marriage is still a marker of better wellbeing. As mentioned earlier, it has been suggested that the deinstitutionalisation of marriage and the growing institutionalisation of cohabitation will result in cohabitation becoming indistinguishable from marriage in couple formation (Cherlin 2004). In the case of child-rearing couples, it seems that this is not yet the case. Thus, it seems that marriage has not lost its special place in the family system; it is perhaps a sign of strong personal commitment.

## 5. Conclusions

A reasonable conclusion from a research point perspective is that treating cohabiting and married couples as belonging to the same subsample when studying parental wellbeing in Finland, as has previously been done, should be avoided. Instead, when studying family health and child development, we need to embrace intersectional diversity in family formation: cohabiting, married and single, same-sex families, and new formations of families in different constellations should be studied as they appear, instead of pooling them into the dichotomy "Married/Single".

This study examined only heterosexual parents, and, thus, based on the results we cannot say whether the same distinction is relevant for same-sex parents. However, according to the commitment framework, it seems probable that this division into subgroups is also relevant for same-sex parents, especially today when same-sex couples have the legal right to marry in Finland. In addition, the movement supporting the legal right to marry for same-sex couples seems to underline the symbolic importance of formal marriage suggested by Cherlin (2004, 2020). Finally, the main aim of this research project is to study different factors related to parenting and child outcomes. Therefore, it is important to emphasise that the aim of the study is not to speak for or against a certain family structure. Instead, primary health care needs to acknowledge diversity and to address the different needs of different families, and to provide support for parenting regardless of family structure. This study confirms that cohabiting couples expecting their first child are in many ways more vulnerable than married couples, and in need of more and perhaps different support.

**Author Contributions:** Conceptualization, M.K. and M.P.; methodology, L.V. and S.R.; formal analysis, L.V. and S.R.; investigation, M.K.; data curation, S.R.; writing—original draft preparation, M.K.; writing—review and editing, S.S., M.P., J.L., J.S., M.S.-B.; project administration, M.K.; funding acquisition, M.K. All authors have read and agreed to the published version of the manuscript.

**Funding:** This research was funded by Signe and Ane Gyllenberg Foundation, nr 4669.

**Institutional Review Board Statement:** The study was conducted in accordance with the Declaration of Helsinki, and approved by the ethical committee of the Finnish National Institute for Health and Wellbeing 2/2011.

**Informed Consent Statement:** Informed consent was obtained from all subjects in the study.

**Data Availability Statement:** The data presented in this study are available on request from the corresponding author. The data are not publicly available due to containing sensitive data on health and wellbeing.

**Acknowledgments:** Open access funding provided by University of Helsinki.

**Conflicts of Interest:** The authors declare no conflict of interest.

## Appendix A

| Mothers | Cohabiting | | | Married | | | | |
|---|---|---|---|---|---|---|---|---|
| | **n** | **M(SD)** | **%** | **n** | **M(SD)** | **%** | **%$t$-test$^2$** | **$\chi^2$** |
| Maternity Clinic | 251 | 4.45 (0.82) | 99.2% | 271 | 4.54 (0.72) | 98.5% | 0.208 | c |
| Pregnancy Group Maternity Clinic | 251 | 3.55 (1.10) | 86.9% | 271 | 3.64 (1.08) | 88.9% | 0.382 | 0.467 |
| Pregnancy Group Other | 251 | 2.64 (1.57) | 21.1% | 270 | 2.77 (1.54) | 21.1% | 0.661 | 0.999 |
| Friends | 251 | 4.57 (0.75) | 98.4% | 271 | 4.71 (0.61) | 98.9% | **0.016** | c |
| Siblings | 250 | 3.89 (1.26) | 73.6% | 271 | 3.87 (1.32) | 70.5% | 0.931 | 0.428 |
| Mother | 251 | 4.35 (0.98) | 96.0% | 271 | 4.43 (0.90) | 95.2% | 0.382 | 0.651 |
| Father | 251 | 3.65 (1.23) | 86.9% | 271 | 3.58 (1.29) | 83.0% | 0.590 | 0.223 |
| Partner | 251 | 4.77 (0.59) | 98.8% | 271 | 4.9 (0.36) | 99.6% | **0.004** | c |
| Internet | 251 | 4.19 (0.86) | 94.8% | 271 | 4.33 (0.86) | 96.3% | 0.068 | 0.407 |
| Facebook | 250 | 2.63 (1.27) | 60% | 271 | 2.69 (1.40) | 55% | 0.677 | 0.247 |
| Reading onl. | 251 | 3.46 (1.11) | 83.3% | 271 | 3.59 (1.11) | 82.3% | 0.230 | 0.767 |
| Discussing onl. | 251 | 2.12 (1.41) | 39.0% | 271 | 2.58 (1.67) | 33.9% | 0.045 | 0.227 |
| Books, magazines | 251 | 3.98 (0.90) | 93.6% | 271 | 3.99 (0.91) | 97.0% | 0.945 | 0.062 |
| TV and radio | 250 | 3.04 (1.13) | 75.6% | 271 | 2.94 (1.24) | 70.8% | 0.388b | 0.222 |
| Professional help | 251 | 3.32 (1.50) | 36.7% | 270 | 3.33 (1.55) | 33.3% | 0.936 | 0.427 |
| Other | 244 | 3.10 (1.62) | 20.5% | 264 | 3.53 (1.60) | 18.6% | 0.186 | 0.583 |

| Father | Cohabiting | | | Married | | | | |
|---|---|---|---|---|---|---|---|---|
| | **n** | **M(SD)** | **%** | **n** | **M(SD)** | **%** | **%*t*-test²** | **$\chi^2$** |
| Maternity Clinic | 91 | 4.08 (1.00) | 83.9% | 139 | 4.20 (0.84) | 95.7% | 0.323 | **0.028** [a] |
| Preg Gr Mat | 91 | 3.55 (1.18) | 82.4% | 139 | 3.68 (1.05) | 91.4% | 0.415 | **0.042** |
| Preg Gr Oth | 91 | 1.88 (1.21) | 35.2% | 139 | 2.30 (1.23) | 30.9% | 0.138 | 0.503 |
| Friends | 91 | 4.08 (0.93) | 91.2% | 139 | 4.02 (0.98) | 94.2% | 0.610 | 0.535 [a] |
| Siblings | 91 | 3.46 (1.43) | 76.9% | 139 | 3.58 (1.18) | 75.5% | 0.550 [b] | 0.810 |
| Mother | 91 | 3.86 (1.20) | 91.2% | 139 | 4.02 (1.02) | 89.2% | 0.302 | 0.787 [a] |
| Father | 91 | 3.51 (1.28) | 86.8% | 139 | 3.66 (1.17) | 82.7% | 0.385 | 0.405 |
| Partner | 91 | 4.56 (0.81) | 96.7% | 139 | 4.77 (0.60) | 97.8% | **0.034** [b] | [c] |
| Internet | 91 | 3.64 (1.13) | 89.0% | 139 | 3.87 (0.99) | 89.9% | 0.137 [b] | 0.824 |
| Facebook | 91 | 1.92 (1.11) | 58.2% | 139 | 1.85 (1.05) | 43.9% | 0.722 | **0.033** |
| Reading onl. | 91 | 2.49 (1.25) | 64.8% | 138 | 2.98 (1.24) | 62.3% | **0.022** | 0.699 |
| Discussing onl. | 91 | 1.49 (0.99) | 40.7% | 138 | 1.39 (0.80) | 37.0% | 0.623 | 0.573 |
| Books, magazines | 91 | 3.41 (1.03) | 80.2% | 139 | 3.76 (0.93) | 88.5% | 0.016 | 0.084 |
| TV and radio | 91 | 2.80 (1.05) | 75.8% | 139 | 2.65 (1.04) | 72.7% | 0.381 | 0.593 |
| Professional help | 91 | 1.90 (1.45) | 31.9% | 139 | 2.48 (1.57) | 37.4% | 0.102 | 0.390 |
| Other | 91 | 1.80 (1.22) | 27.5% | 139 | 4.20 (0.84) | 26.6% | 0.195 | 0.887 |

Reading onl. = reading online boards, Discussing onl. = discussing on online boards. [a] Yates' Correction, [b] Equal variances not assumed, [c] Significance not reported due to too small (<5) expected count in multiple cells.

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
