# Peer review of "Married and Cohabiting Finnish First-Time Parents: Differences in Wellbeing, Social Support and Infant Health"

_socsci, doi:10.3390/socsci11040181_

Round 1

Reviewer 1 Report

I could not copy my review into the box.

Author Response

Thank you so much for these encouraging words!

Reviewer 2 Report

The manuscript focuses on potential differences in parental wellbeing, infant health and institutional or social support according to the marital status of Finnish first-time parents. Survey data from both mothers and fathers is used, and some interesting differences are found between cohabiting and married parents(-to-be). The authors have identified a clear gap in prior research: the lack of differentiating between cohabiting and married parents(-to-be) when studying parental wellbeing in a country context (Finland) with a majority of firstborns born outside marriage, and in which cohabitation has long been a normalized form of relationship.

General comments:

The paper makes an interesting contribution to the research literature on the differences between cohabitation and marriage, the wellbeing and health outcomes of these two types of family forms, and, more generally, even on what marriage and cohabitation might represent in today’s liberal societies. It could thus be of interest for scholars interested in parental and child wellbeing but also in more theoretical conceptualizations of families and close relationships. The text is written in a clear style, with only minor spelling checks needed, and the manuscript is mostly clearly structured under relevant subtitles. The discussion and conclusions are interesting and consistent with the empirical results, and the theoretical/conceptual approach is certainly a strength of the paper.

However, the paper could be improved in some areas. The weaknesses relate mainly to the description of the method, presentation of the empirical results, and the consistency of the research questions and the results.

The research questions as such are well defined, and the data and methods chosen are suitable for answering them. It is therefore confusing that, when presenting the results, the authors include a new variable (the baby being planned or not) that was not introduced in the research questions or in the presentation of the analysed variables. I would recommend the authors either to stick to the research questions and remove the variable ‘baby as planned’ from the analysis, or, if they want to add other variables to the analysis, to include them in the research questions with a clear rationale and a presentation of the variables in the method section.

The section 2 on data and methods lacks several important information. First, the description of the data should include information on when the data was collected (year). Obviously, it is not possible to define response rates, but even a rough estimation of how many families were contacted through maternity clinics in the studied municipalities, and thus, how large a share of the families did or did not take part in the study would help estimate the coverage of the data.

Second, a list of the independent/control variables should be provided, since now only the outcome variables and the main independent variable, marital status, are listed.

Third and most importantly, the paper lacks a specific description of the methods of analysis used (e.g. means, t-test, OLS and logistic regression). The methods become obvious only when reading the section on results, but it is quite laborious to the reader to search the methods used for analysing the different outcome variables in this part of the text. Moreover, no description of how the different (outcome and control) variables were introduced to the models is included. The methods should preferably be described in detail in the same section with the data and variables in order to enable the reader to evaluate the methodological accuracy of the results.

Additionally and for the same reason, it would be important to add tables, under the section 3 on results, which would present the findings based on OLS and logistic regression. The analysis, and the findings based on them, are now only presented as figures in the text. As some of the figures represent associations between two variables without and others with controlling for other variables, it would be easier to follow and evaluate the findings with tables including these different types of models. In the text too, the results of the logistic/OLS regressions that do or do not control for all the independent variables should be clearly differentiated. This is not always clear to the reader (e.g., lines 211-213). It could be less confusing to the reader if only the ‘final’ results including all the control variables (gender, age, education?) were presented in the text.

To help the reader in following how the results were achieved at, the section 3 on results could be more clearly organized around the different outcome variables and their associations with marital status. As a reader, it is hard to understand the logic and the story line of this section, as currently it is a mix of results presenting the outcome variables in a confused order but also other variables (which are sometimes used as independent and sometimes as dependent variables). Especially the part under heading 3.1 would benefit from a clearer structure: for example, first the results on depressive symptoms, then on relationship satisfaction, etc. Findings that do not answer the research questions and do not correspond to the heading 3.1 ‘Differences in parental wellbeing and infant outcome’, such as comparisons of educational levels or age by marital status (e.g. lines 199-206), could be presented in the beginning of the results section 3, not under 3.1. It would help the reader if the results would focus on answering the research questions, and all the other information would be moved elsewhere or removed.

When analysing social support, independent variables other than marital status such as age or education are not controlled for. This is an acceptable solution but could be given an explanation and a thought in the discussion. For example, could educational level (partly) explain why cohabiting fathers used the services of maternity clinics less typically than married fathers?

In addition, the paper would benefit from a short review of previous research (in Finland or in other countries) on marital status and wellbeing outcomes already in the introduction. The authors do present previous research on associations between the wellbeing variables and child wellbeing outcomes, and by this they seem to explain why the chosen outcome variables are relevant. However, only one study is mentioned, in the introduction, on associations between marital status and wellbeing outcomes, which would argue for choosing marital status as the main independent variable. Some research findings are referred to in the discussion part of the paper, but I would suggest including these prior findings in the introduction. Moreover, some of the references are quite old. If little recent research is found on this area, this could be explained too. I also wonder why statistics for Finland are from year 2015. Is this maybe the year the data was collected? If yes, the explanation for using seven years old figures could be added; if not, the information could be updated if possible.

Finally, in the beginning of the discussion (lines 296-307), the authors nicely summarize the main results. However, only the associations that were confirmed (how marital status was associated with depression, relationship satisfaction and social support) are summarized, while nothing is said about the associations that did not exist, according to the findings. I would suggest including a short summary also on the ‘lacking’ associations. As findings, they are just as important than the associations that were found in the analysis. Why marital status was not associated with all the studied outcomes could be (shortly) reflected on.

A possible selection bias of the respondents (they had higher educational level than on average, how about other possible groups that are under-represented in the data?) could also be mentioned and discussed as a weakness of the study.

Specific comments

Lines 36-45: The main point of the paragraph is a bit unclear, and it is hard to interpret how the paragraph connects to the previous and to the following paragraphs. As this is the introduction to the paper, please pay special attention to helping the reader to follow the argumentation.

Line 108 (and other relevant parts): The data was collected in ‘communities’ – do you mean municipalities (maternity clinics as municipal services?)? ‘Communities’ remains vague.

Line 239, table 3. The title of the table could be more informative, including information on what is compared and by which variable (like in table 4), for example ‘Parental wellbeing by marital status and by gender’.

Line 249, table 4. I assume the title should be ‘Infant outcome by marital status’ (not ‘family type’). In the same table 4, why is the value for p missing for ILLNESS?

Lines 260-265: Please clarify what is meant by ‘We removed them from the data …’. Do you mean that you 1) first looked at the use of the different types of support by marital status, so in this phase those who did not use any support were still included, 2) then removed the 'non-users' and analysed how the users rated the support?

Line 270: What is coaching service? This term would require a short explanation (e.g. in the introduction where maternity clinics are introduced).

Tables 5 and 6: The captions are not in the right columns. Please explain M. coaching and O. coaching.

Line 322: An uncommon word for choices, ‘liquid’, is used. The theoretical context implies that this might refer to the conceptualizations of Zygmynt Baumann. If so, the reference could be included.

Line 340: I think it should read ‘…some of the cohabiting couples in our sample…’ (This is an interesting discussion. It implies that there would be more variation among cohabiting couples in the dimensions of wellbeing than among married couples. Did you check this and if so, did your results support this conclusion? So some of the cohabiting couples might be as committed - and as satisfied and not depressed - than the married couples, but the group would also include romantically uncommitted couples which is why the group differs from the group of married parents-to-be.)

Line 359: ‘The questionnaire reflected…’ could be reformulated (as the questionnaire was drafted by researchers and could not yet reflect any results as such).

Line 373: Following the families until the child turns two is not actually a strength of this analysis (although it could be a strength of the wider research project). This strength could be removed or reformulated.

Line 386: Please be more specific with this recommendation: it might be true when studying parental wellbeing, while ‘studying parenthood’ is quite vague (as parenthood studies include a wide variety of themes for which marital status might be more or less relevant).

Reviewer 3 Report

The manuscript is well written. However the author(s) need to address several minor issues:-

Material and method

The author(s) need to carried out the analysis to determine the regression model (OLS or Logistic( is fit to ensure the validity of the results.

Discussion

More literature on the similar studies should be included to "enrich" the content of the discussion

Author Response

Thank you for your encouraging words!

  1. Thank you for this valuable suggestion. We have now presented the results of OLS and Logistic Regression in Tables and structured the Result section differently. We hope this is satisfying.
  2. We have added studies about marital status related to infant outcome/ child development/relationship quality in the Introduction, and in addition enriched the discussion.

Round 2

Reviewer 2 Report

The manuscript has been remarkably improved: the research questions and the results presented are now consistent, and the results are much easier to follow. However, I still have some minor suggestions that are listed below. Especially some small clarifications to the methods, variables and tables would make it easier to understand and evaluate the use of methods in this study.

The dependent variable 'baby being planned for', now included in the research questions, should be added and shortly explained also under the section 2 'Materials and methods' together with other dependent variables.

The section 3.1 now starts with the results of the OLS and logistic regressions. This is a nice way of highlighting the 'main' results of the article based on the analysis where certain variables (gender, age, education) were controlled for. However, it now remains unclear to the reader why only EPDS, IMS, baby planned and smoking were chosen to be included in the regressions. The results of the table 5 seem to offer an explanation (all the dependent variables were first analysed in respect to marital status, and then only the variables that proved to be associated with marital status were chosen for further analysis with control variables?) but the logic of the analysis could be explained more clearly (if not in the section 2, then at the beginning of 3.1).

Tables 3 and 4: The tables make understanding the results easier but still, it is not clear how the independent variables (marital status, age, education, gender) were coded, that is, which is the reference category. I would still suggest explaining the use of these independent variables under the heading 2 'Materials and methods' (after having explained the dependent variables), or specifying the categories of the variables (with the reference category visible) in the tables.

At least some of the dependent variables in the section 2 'Materials and methods' lack a description of how they were coded to conduct the analysis, e.g. baby being planned for, smoking (what are the categories and which one is used as a reference category for logistic regression). The scale for subjective health is also missing; it could be added either under 'Materials and methods' or to the note under Table 5.

Tables 3 and 4, notes on lines 205-206 and 227-228: explanation for *** is missing.

Line 271: 'Coaching service' is still mentioned in the text although changed to 'pregnancy groups' in the tables.
